# Direct radiative effect of dust-pollution interactions

Klaus Klingmüller[1], Jos Lelieveld[1,2], Vlassis A. Karydis[1,3], and Georgiy L. Stenchikov[4]

[1]Max Planck Institute for Chemistry, Hahn-Meitner-Weg 1, 55128 Mainz, Germany

[2]The Cyprus Institute, P.O. Box 27456, 1645 Nicosia, Cyprus

[3]Forschungszentrum Jülich GmbH, IEK-8, 52425 Jülich, Germany

[4]King Abdullah University of Science and Technology, Thuwal 23955-6900, Saudi Arabia

**Correspondence:** Klaus Klingmüller (k.klingmueller@mpic.de)

**Abstract.** The chemical ageing of aeolian dust, through interactions with air pollution, affects the optical and hygroscopic properties of the mineral particles and hence their atmospheric residence time and climate forcing. Conversely, the chemical composition of the dust particles and their role as coagulation partners impact the abundance of particulate air pollution. This results in a change in the aerosol direct radiative effect that we interpret as an anthropogenic radiative forcing associated with mineral dust-pollution interactions. Using the ECHAM/MESSy atmospheric chemistry climate model (EMAC), which combines the Modular Earth Submodel System (MESSy) with the European Centre/Hamburg (ECHAM) climate model, including a detailed parametrisation of ageing processes and an emission scheme accounting for the chemical composition of desert soils, we study the direct radiative forcing globally and regionally considering solar and terrestrial radiation. Our results indicate positive and negative forcings, depending on the region. The predominantly negative forcing at the top of the atmosphere over large parts of the dust belt, from West Africa to East Asia, attains a maximum of about $-2 \ \mathrm{W/m^2}$ south of the Sahel, in contrast to a positive forcing over India. Globally averaged, these forcings partially counterbalance, resulting in a net negative forcing of $-0.05 \ \mathrm{W/m^2}$, which nevertheless represents a considerable fraction (40 %) of the total dust forcing.

## 1 Introduction

Atmospheric aerosols play an important role in the climate system by affecting radiative transfer and thus the planet's energy budget, both directly by scattering and absorption and indirectly via its impact on cloud formation (IPCC, 2014). Furthermore,

fine particulate matter can be a human health hazard and is a major cause of morbidity and mortality globally (Lelieveld et al., 2015).

Aerosols originate both from natural and anthropogenic sources, the former being mostly mineral dust, sea salt and emissions from naturally ignited fires. Mineral dust is the dominant aerosol component by mass and natural sources are responsible for most of its atmospheric load, even though about 25 % may be from human-made sources (Ginoux et al., 2012). The natural sources provide an inevitable background level of atmospheric particulate matter, while studies of the human impact on climate and air pollution commonly focus on aerosol from anthropogenic sources. However, within the atmosphere natural and anthropogenic aerosols are mixed and interact, and therefore should not be considered separately.

In the presence of anthropogenic pollution, gaseous compounds, notably acids, condense on the mineral dust particles (Karydis et al., 2011). The consequent interactions are dubbed chemical ageing, converting the initially hydrophobic dust particles into hydrophilic ones (Karydis et al., 2017), leading to the hygroscopic growth of the particles with implications for their optical properties and the rate of deposition (Levin et al., 1996; Abdelkader et al., 2015, 2017). The dust particles also serve as coagulation partners for particulate anthropogenic pollution. Moreover, the chemical composition of the dust particles affects the chemical properties of the aerosol mixture (Karydis et al., 2016) and hence the hygroscopic and optical properties as well as the atmospheric residence time of both the natural and anthropogenic components. In view of emerging economies with growing population and increasing emissions from industry, energy production and transport in dust affected regions such as northern Africa, the Middle East and large parts of Asia, the importance of these effects is ever-increasing (Osipov et al., 2015; Osipov and Stenchikov, 2018).

In the present study we analyse the impact of mineral dust interactions with anthropogenic air pollution on radiative transfer using the ECHAM/MESSy chemistry climate model (EMAC) (Jöckel et al., 2005, 2010). EMAC combines the Modular Earth Submodel System (MESSy) with the European Centre/Hamburg (ECHAM) climate model which is originally based on the weather forecasting model of the European Centre for Medium-Range Weather Forecasts (ECMWF). Here we focus on the direct radiative effects while not considering aerosol cloud coupling, and ignoring radiative feedbacks on the climate system. Both aspects, of which especially the former influences radiative forcing, will be considered in a separate study based on climate model simulations that account for atmosphere-ocean coupling.

Our main goal is to understand how the mineral dust in the present day atmosphere differs from dust under natural conditions and in particular to evaluate the implications on the global and regional radiative transfer, which is the focus of the present study. Accordingly we choose the methodology described in the following section which, in contrast to previous studies (Abdelkader et al., 2015, 2017), does not alter the chemical ageing mechanism but rather the emissions. The impact of dust on pollution is

crucial and therefore considered by our approach on an equal footing with the impact of pollution on dust. A technical objective of this study is to assess the error introduced in climate models if mineral dust and anthropogenic pollution are assumed to be coexisting without any interaction and thereby to point out the importance of taking the interactions into account.

The article is structured as follows: In section 2 we present our methodology including the model setup-up. The effects of dust-pollution interactions on the aerosol burdens and correspondingly the optical properties are analysed in section 3, the resulting impacts on radiative transfer and atmospheric heating in section 4. Conclusions are drawn in section 5.

## 2 Methodology

We used the EMAC model version and configuration described by Klingmüller et al. (2018), which was shown to yield realistic results of aerosol optical properties globally (see Fig. S1 in the supplement). This EMAC version combines ECHAM 5.3.02 and MESSy 2.52 and is configured to use the horizontal resolution T106 and 31 vertical levels. The grid spacing of the Gaussian T106 grid, $1.125°$ along latitudes and about $1.121°$ along longitudes, at the equator corresponds to virtually quadratic cells with around 125 km edge length. The following MESSy submodels have been enabled: AEROPT, AIRSEA, CLOUD, CLOUDOPT, CONVECT, CVTRANS, DDEP, GMXE, JVAL, LNOX, MECCA, OFFEMIS, ONEMIS, ORBIT, ORACLE, PTRAC, RAD, SCAV, SEDI, SURFACE, TNUDGE, TROPOP. Descriptions of each submodel and further references can be found online in the MESSy submodel list (MESSy 2018). The model dynamics above the boundary layer are nudged to meteorological analyses of the European Centre for Medium-Range Weather Forecasts (ECMWF), and the aerosol radiative effect on the dynamics is computed using the extinction, single scattering albedo and asymmetry factor from the Tanre aerosol climatology (Tanre et al., 1984). The aerosol radiative coupling to the meteorology, and that between aerosols and clouds have been disabled to exclude higher order effects such as feedbacks by precipitation and evaporation changes and to focus on the direct radiative forcing. The CMIP5 RCP4.5 (Coupled Model Intercomparison Project Phase 5 Representative Concentration Pathway 4.5) (Clarke et al., 2007), GFEDv3.1 (Global Fire Emissions Database) (Randerson et al., 2013) and AeroCom (Aerosol Comparisons between Observations and Models) (Dentener et al., 2006) databases provide anthropogenic, biomass burning and sea salt emissions, respectively.

The EMAC model considers dust ageing by condensation of soluble compounds, ionization, hydrolysis and the associated water uptake. The model representation of the hygroscopic variations has been evaluated against field measurements by Metzger et al. (2016). Abdelkader et al. (2017) evaluated the chemical ageing during the transatlantic dust transport using ground based AERONET (Aerosol Robotic Network) observations and satellite retrievals from MODIS (Moderate Resolution Imaging Spectroradiometer) and CALIPSO (Cloud-Aerosol Lidar and Infrared Pathfinder Satellite Observations). Consistent results

where also reported by other studies (e.g., Abdelkader et al., 2015; Klingmüller et al., 2018; Brühl et al., 2018). The relevant submodels include the Global Modal Aerosol Extension (GMXE) (Pringle et al., 2010a, b), which simulates the aerosol microphysics considering four soluble (nucleation, Aitken, accumulation, coarse) and three insoluble modes (Aitken, accumulation, coarse). The size distribution of each mode is represented by a log-normal distribution with fixed geometric standard deviation ($\sigma_g = 2$ for the two coarse modes, $\sigma_g = 1.59$ for all others), fixed dry radius boundaries between the nucleation, Aitken, accumulation and coarse modes (6 nm, 60 nm and 1 $\mu$m) and variable mean radius. GMXE employs ISORROPIA II (Fountoukis and Nenes, 2007) or EQSAM4clim (Equilibrium Simplified Aerosol Model V4 for climate simulations) (Metzger et al., 2016) for the gas-aerosol partitioning (here we use the former). The amount of gas kinetically able to condense is calculated assuming diffusion limited condensation using the accommodation coefficients in Table 2 before ISORROPIA II re-distributes the mass between gas and aerosol phase to obtain the final amount of condensed material (Pringle et al., 2010a, b). The ORACLE (Organic Aerosol Composition and Evolution) submodel comprehensively describes organic aerosols (Tsimpidi et al., 2014). A detailed simulation of the gas phase chemistry is performed by the Module Efficiently Calculating the Chemistry of the Atmosphere (MECCA) (Sander et al., 2011).

Aerosol optical properties are calculated by the AEROPT (AERosol OPTical properties) submodel (Lauer et al., 2007; Pozzer et al., 2012; Klingmüller et al., 2014), which assumes the aerosol components within each mode to be well mixed in spherical particles with volume averaged refractive index. The refractive indices AEROPT considers for the individual components are compiled from the OPAC 3.1 database (Hess et al., 1998) (black carbon, mineral dust), the HITRAN 2004 database (Rothman et al., 2005) (organic carbon, sea salt, ammonium sulphate, water), Kirchstetter et al. (2004) (organic carbon for $\lambda < 0.7\mu$m) and additional mineral dust values for $\lambda > 2.5\mu$m (see Klingmüller et al., 2014). The full dataset is specified in the supplement (Fig. S25, Tables S1, S2). The imaginary part of the dust refractive index used here attains a minimum of $4 \cdot 10^{-3}$ at visible and near-infrared wavelengths. AERONET retrievals might yield lower values, e.g., Gómez-Amo et al. (2017) report 2.5 to $2.8 \cdot 10^{-3}$ over Spain, however due to ageing the modelled dust is usually mixed with other components and especially water so that the effective imaginary refractive index of the entire particles is lower than the value assumed for pure dust.

The dust emission scheme is part of the online emission submodel ONEMIS (Kerkweg et al., 2006b). We use the dust emission scheme presented by Klingmüller et al. (2018) which is based on Astitha et al. (2012) and differentiates the $Ca^{++}$, $K^+$, $Mg^{++}$ and $Na^+$ fractions in mineral particles originating from different deserts (Karydis et al., 2016). The majority of the mineral dust mass is emitted into the coarse mode (approximately 89 %), the remainder into the accumulation mode. Freshly emitted mineral dust is assumed to be hydrophobic and thus emitted into the insoluble modes. Only after condensation of

sufficient soluble material to cover the particles with 10 monolayers or by coagulation with soluble particles, initially insoluble particles are transferred to the soluble modes (Vignati et al., 2004; Stier et al., 2005; Pringle et al., 2010a, b). The removal by wet deposition is simulated by the scavenging submodel SCAV (Tost et al., 2006), dry deposition and sedimentation by the submodels DDEP and SEDI (Kerkweg et al., 2006a).

Our analysis covers the meteorological year 2011. Four simulations with varied emission setups (Table 1) are used to derive the instantaneous forcing by the interaction of mineral dust and anthropogenic pollution: one simulation considering all emissions (simulation 1), the same simulation but without dust emissions (simulation 2), a simulation with only natural emissions (simulation 3) and the corresponding simulation without dust emissions (simulations 4). For the natural emission setups we omit the CMIP5 anthropogenic emissions and reduce the GFED biomass burning emissions by 90 % (Levine, 2014). All dust

emissions are considered to be natural, hence the anthropogenic impacts of land use and climate change on the dust emissions (Klingmüller et al., 2016) are excluded from our analysis. The contribution of dust-pollution interactions to the total aerosol radiative forcing can be calculated from the aerosol forcings $F_{1...4}$ from the four simulations 1 to 4 by evaluating

$$\Delta F = (F_1 - F_2) - (F_3 - F_4), \tag{1}$$

the difference of the dust forcing with full emissions $F_1 - F_2$ and the dust forcing with only natural emissions $F_3 - F_4$.

Analogously, we define the dust-pollution interaction effect on aerosol optical depth (AOD), atmospheric heating rates and aerosol particle burdens.

Note that the right hand side of Eq. 1 is symmetric regarding the exchange of dust and anthropogenic emissions, i.e., it considers the effect of the pollution on dust in the same way as the effect of dust on pollution by comparing the forcing of anthropogenic pollution in the presence of dust ($F_1 - F_3$) with its forcing in a dust free scenario ($F_2 - F_4$). Unlike the pollution

free scenario, the globally dust free scenario has no counterpart in the real world, nevertheless this interpretation would be suitable to evaluate the regional impact of dust events or new dust sources on pollution. It can be instructive to expand the RHS of Eq. (1) to the difference of the combined dust and pollution forcing $F_1 - F_4$ and the sum of the forcing of only pollution $F_2 - F_4$ and only dust $F_3 - F_4$,

$$\Delta F = (F_1 - F_4) - \big((F_2 - F_4) + (F_3 - F_4)\big). \tag{2}$$

Due to clouds, radiative forcings strongly vary over time and accordingly their temporal averages are associated with substantial statistical uncertainty even for relatively long averaging intervals. The computation of $\Delta F$ can be challenging if this

uncertainty is uncorrelated between the individual terms $F_{1...4}$. If the aerosol-cloud and radiative coupling to meteorology are taken into account so that the cloud cover of the four simulations is no longer identical it is essential not to use total fluxes on the RHS of Eq. 1 but to calculate the aerosol forcing as the difference between fluxes computed by two simultaneous radiative transfer computations, one with and one without considering the aerosol but both with identical cloud effect (Ghan et al., 2012; Dietmüller et al., 2016). This eliminates most of the cloud related statistical noise and drastically reduces the length of the averaging period which is required to obtain significant results. Nevertheless, in the present study the cloud cover is identical for all four simulation because the different emission setups do not affect the meteorology.

To estimate the remaining statistical uncertainty, we split the time series of daily averages into $n$ sub-samples, each consisting of only every $n$-th daily value. As long as the choice of $n$ is not too large, this ensures that each sub-sample is unbiased by seasonality. We consider the random terms of the sub-samples to be largely uncorrelated, which allows the computation of the statistical uncertainty as standard error of the mean (SEM) of the results from all sub-samples. To obtain approximate uncertainty estimates, we use $n = 5$ for annual and $n = 7$ for seasonal analyses, which are small numbers with regard to the SEM calculation but ensure representative subsets and are factors of the number of days per year and season, respectively. We use the resulting uncertainty estimate $\sigma$ to apply a significance threshold of $2\sigma$ to our results. For our purpose, the one year simulation period turns out to be sufficient to produce significant results.

The interannual variation is estimated based on a 10 year simulation at lower T63 resolution (about $1.9°$) which yields coefficients of variation (CV) below 10 % for our main results. We conclude that while the interannual variation is not negligible it does not substantially affect our results. Other uncertainties such as biases in the parametrisations and emission inventories are likely more relevant. This especially applies to results which integrate partially compensating regional positive and negative contributions such as the change of the global top of the atmosphere (TOA) forcing by the dust-pollution interactions. EMAC yields similar results for different commonly used parametrisations of the optical properties of internally mixed particles (Klingmüller et al., 2014) so that the associated error is small as long as the particles are spherical which is the case after hygroscopic growth. Considering non-spherical particles could improve the parametrisation for freshly emitted, dry mineral dust, but no substantial corrections are to be expected (Koepke et al., 2015).

The difference between the global direct aerosol radiative forcings in simulation 2 and simulation 4 yields an anthropogenic aerosol forcing of $-0.61 \text{ W/m}^2$ at the TOA (see Figs. S2 to S5 in the supplement), consistent with the estimate of the aerosol-radiation interaction effective radiative forcing (ERF) of $-0.45$ ($-0.95$ to $0.05$) $\text{W/m}^2$ indicated by IPCC (2014).

The global dust radiative forcing excluding the effect of dust-pollution interaction can be calculated as difference between the aerosol forcings in simulation 3 and simulation 4. At the TOA the net forcing amounts to $-0.08 \text{ W/m}^2$, comprising the solar

radiation forcing of $-0.16\,\mathrm{W/m^2}$ and the terrestrial radiation forcing of $0.09\,\mathrm{W/m^2}$ (see Figs. S6 to S9 in the supplement). The net forcing is less negative than the $-0.14\,\mathrm{W/m^2}$ reported by (Bangalath and Stenchikov, 2015), but well within the range of $-0.48$ to $0.20\,\mathrm{W/m^2}$ estimated by Kok et al. (2017) and the wide spread of forcings from different models (Fig. S10 in the supplement, Yue et al. (2010), Table 1). The total yearly mineral dust emission and deposition rate is $1.31\mathrm{Gt/yr}$, slightly higher than the AeroCom estimate of $1.1\mathrm{Gt/yr}$ (Huneeus et al., 2011) and within the range $1.7$ $(1.0$ to $2.7)\mathrm{Gt/yr}$ provided by Kok et al. (2017). The global average 550 nm dust AOD of $0.021$ is comparable to the AeroCom median of $0.023$ (Huneeus et al., 2011) but lower than the $0.03\pm0.005$ reported by Ridley et al. (2016). The relatively small dust AOD and forcings in the present study compared to previous work suggest that our estimates for the radiative effect of dust-pollution interaction which predominantly affects the solar spectrum may be considered as conservative.

## 3   Aerosol burdens and optical properties

The condensation of soluble compounds, their reaction and the consequent hygroscopic growth increase the size of the dust particles and thereby their dry deposition velocity and the efficiency of in- and below-cloud scavenging. Figure 1 (top) shows that the anthropogenic dust ageing significantly reduces the dust and hence the coarse mode annual mean mass burden throughout most dust affected regions (the right column of Fig. S1 in the supplement highlights the relevant the regions). The only notable exception is over the western Atlantic Ocean. Here, the mineral dust moderates the reduction of sea salt and the associated water by anthropogenic pollution.

The effect on the accumulation mode aerosol burden, being most relevant for the AOD and hence radiative transfer, is more complex because not only the effect of pollution on dust but also the effect of dust on accumulation mode pollution is relevant. Generally, coarse mineral dust particles transfer aerosol mass from the accumulation mode to the coarse mode by coagulation. Therefore the interaction effect on the burden of no-dust aerosol components tends to be negative for the accumulation mode but positive for the coarse mode. As Figure S11 in the supplement shows, this applies to the black carbon (BC), sea salt and water burdens. Aerosol components which interact with the mineral cations of the dust behave differently. For instance, in the full emission simulation 1, unlike the dust free simulation 2, ammonium is driven out of the aerosol phase by the mineral cations (Metzger et al., 2006), which results in reduced aerosol ammonium burdens but increased gas phase ammonia burdens (Fig. S12 in the supplement). This predominantly affects the accumulation mode which contains most ammonium. Conversely, the aerosol nitrate burdens are enhanced through the interaction of mineral cations with gas phase nitric acid. In contrast, similarly to non-ionic components such as BC, aerosol sulphate is transferred from the accumulation mode to the coarse mode through coagulation in the presence of coarse dust particles.

The changes of the accumulation mode composition reduce the hygroscopicity, the amount of accumulation mode water and the AOD. This also prolongs the atmospheric residence time of accumulation mode pollution particles in the full emission simulation 1 compared to the dust free simulation 2, increasing the burden. In comparison with the pollution free simulation 3, the accumulation mode burden is enhanced by the reduced coagulation with coarse mode dust particles which are more effi-

5 ciently removed in the presence of pollution. Therefore the interaction effect on the accumulation mode dust burden is positive (Fig. S11 in the supplement). In our simulation, the effects that increase the accumulation mode burden generally outweigh the decrease due to more efficient deposition of accumulation mode dust particles. As shown in Fig. 1 (bottom), the interaction of anthropogenic pollution and dust results in an increased annual mean accumulation mode burden over most regions. The strongest increase we obtain south of the eastern part of the Sahel (point A in Fig. 1). The burden changes of the main aerosol

components over this point which are exemplary for the whole region (region A, cp. Table 3) are analysed in the upper half of Fig. 2. Only over some regions, most notably over Tibet (region B in Fig. 1), the decreased amount of accumulation mode water results in a decreased total accumulation mode burden (lower half of Fig. 2).

The net depleting effect of dust-pollution interactions on the coarse mode aerosol burden reduces the coarse mode contribution to the AOD, but the total AOD in the solar spectrum is dominated by the accumulation mode which is enhanced.

Indeed, the annual mean effect on the AOD distribution depicted at the top of Fig. 3 clearly resembles that of the effect on the accumulation mode shown at the bottom of Fig. 1 (Fig. 3 shows the AOD for the EMAC shortwave band from 250 to 690 nm including the visible wavelengths, the effect on the 550 nm AOD is practically identical, see Fig. S14 in the supplement).

The effect on the aerosol absorption optical depth (AAOD) shown at the bottom of Fig. 3 is slightly negative, due to the higher reflectance and more efficient removal of aged hygroscopic coarse mode dust and the transfer of absorbing components

from accumulation to coarse mode by coagulation. Only south of the Sahel, where the Saharan dust mixes with biomass burning pollution, the AAOD is increased due to the strong AOD increase.

The generally reduced absorption by mineral dust interacting with pollution is also reflected in larger single scattering albedos (SSA) over dust dominated regions in simulation 1 compared to simulation 3 without anthropogenic emissions. Fig. 4 shows the annual mean difference of the SSA in both simulations. The SSA values have been averaged over the vertical

levels weighted with the extinction, corresponding to using $SSA = 1 - AAOD/AOD$. Although carbonaceous components of the anthropogenic pollution reduce the SSA over the remaining globe, over the dust belt the SSA increases by up to 0.01. Together with the uncertainty of the refractive index of mineral dust, neglecting this SSA increase might be responsible for an overestimation of the atmospheric heating by dust (Balkanski et al., 2007). In the terrestrial spectrum aerosol particles are strongly absorbing corresponding to very small SSA values which are approximated by zero in the terrestrial radiative transfer

code. Therefore, unlike the solar radiation, the terrestrial radiation is affected by the dust-pollution interaction only via the modified extinction.

## 4 Radiative forcings and heating rates

The increased AOD and decreased solar radiation absorption due to dust-pollution interactions result in a predominantly negative instantaneous direct top of the atmosphere (TOA) forcing, illustrated in Fig. 5 (top). The consequent climate cooling tendency affects large parts of the dust belt, from West Africa to East Asia, and attains an annual average of about $-2 \ \mathrm{W/m^2}$ south of the Sahel. Positive forcings occur over Asia, exceeding $0.5 \ \mathrm{W/m^2}$ over India.

The distribution of the bottom of the atmosphere (BOA) forcing (Fig. 5, bottom) is similar to that of the TOA forcing, with an annual mean cooling maximum south of the Sahel up to about $-2.5 \ \mathrm{W/m^2}$ and a warming maximum over the Indo-Gangetic Plain exceeding $1 \ \mathrm{W/m^2}$.

Consequently, the net atmospheric forcing is not very large, consistent with the moderate effect on the AAOD, but significant (Fig. 5, centre), and depending on the region can be either negative or positive. The largest region with atmospheric cooling extends from the Sahara over the Middle East to India, reaching an annual mean of $-0.8 \ \mathrm{W/m^2}$ over the Arabian Peninsula. Also over the equatorial Atlantic the dust-pollution interactions result in weak but significant atmospheric cooling. In contrast, south of the Sahel the AAOD increase (see Fig. 3) results in a positive atmospheric forcing up to $0.5 \ \mathrm{W/m^2}$. Over extensive regions in Asia, the forcing is positive as well, but mostly below $0.2 \ \mathrm{W/m^2}$.

The TOA and BOA forcings are dominated by the effect on solar radiation (shortwave, SW, Fig. S17 in the supplement). The effect on the terrestrial radiation (longwave, LW, Fig. S18 in the supplement) yields a forcing which is one order of magnitude smaller than the SW forcing, both globally and regionally. For the atmospheric forcing, the LW contribution is more relevant and depending on the region partially compensates or enhances the SW forcing. For example, the SW heating south of the Sahel and the SW cooling west of the Red Sea are reduced by about 30 %, whereas over the Arabian Peninsula the cooling is enhanced by about 10 %.

Through atmospheric heating and cooling dust-pollution interactions may impact regional atmospheric dynamics. In Fig. 6 we analyse the heating rates in the main regions with negative (regions 1 and 2) and positive (regions 3 and 4) annual mean atmospheric forcing. Over the largest region with net atmospheric cooling, extending from the Sahara over the Middle East to India (region 1), the heating rates show little seasonal variation with a persistent cooling, which reaches a summertime average of $-0.05 \ \mathrm{K/day}$ over the Arabian Peninsula, with a minimum during winter. Similarly, the heating over the largest region with

atmospheric warming, extending from the Sahel to the Congo Basin (region 4), decreases during winter when it turns negative below 3000 m altitude.

In contrast, over regions 2 and 3 the annual average cooling and heating are largest during one season: over the equatorial Atlantic Ocean (region 2) the strongest cooling occurs during winter. Likewise, the heating over Asia is predominant during summer.

Generally, the heating takes place at higher altitudes than the cooling, at times simultaneously, thus stabilising the atmosphere, which is further intensified by the predominantly negative BOA forcing that cools the surface.

When globally averaged, the regionally positive and negative forcings partially counterbalance. Nevertheless, the net annual average global forcing at the TOA is $-0.05$ W/m$^2$, representing a considerable fraction of the total dust forcing: Fig. 7 compares this forcing with that of the dust when neglecting dust-pollution interactions, i.e., the dust forcing in the pollution free scenario $F_{dust} = F_3 - F_4$ which amounts to $-0.08$ W/m$^2$ (SW: $-0.16$ W/m$^2$, LW: $0.09$ W/m$^2$), and the dust forcing including the pollution effects, $F_{dust} = F_1 - F_2$, of $-0.13$ W/m$^2$ (SW: $-0.22$ W/m$^2$, LW: $0.09$ W/m$^2$). It is therefore recommended to take the interactions of dust and anthropogenic pollution into account when assessing the dust radiative forcing as they significantly enhance the net global climate cooling effect of mineral dust.

## 5  Conclusions

The physicochemical interactions of mineral dust with air pollution significantly affect the optical properties, hygroscopicity and atmospheric residence time of dust as well as anthropogenic aerosol particles. We interpret the resulting effect on the radiative transfer as an anthropogenic climate forcing linked to mineral dust, even though most of the dust itself is emitted from natural sources.

Exposed to anthropogenic pollution, insoluble mineral dust particles are turned hygroscopic by chemical ageing. The subsequent hygroscopic growth increases the efficiency of scavenging and deposition, thereby reducing the atmospheric residence time and burden of coarse dust particles. This reduces the coagulation rate of accumulation mode dust particles, increasing the corresponding burden. Other aerosol components are generally transferred from the accumulation to the coarse mode by coagulating with coarse dust particles. The interaction of ionic aerosol components with the mineral cations within the dust particles interferes with this process, driving cations such as ammonium into the gas phase or enhancing the aerosol mass of anions such as nitrate. The combined effect on the aerosol optical properties is dominated by an AOD increase caused by the enhanced accumulation mode dust burden and a decreasing AAOD due to the modified aerosol composition.

The resulting climate forcings are non-uniformly spatially distributed with regionally large positive and negative values. The predominantly negative forcing at the top of the atmosphere over large parts of the dust belt, from West Africa to East Asia, attains a maximum of about $-2$ W/m$^2$ south of the Sahel, in contrast to a positive forcing over India. The surface forcing attains an annual mean of $-2.5$ W/m$^2$ south of the Sahel, in contrast to a mean positive forcing of $1$ W/m$^2$ over the Indo-Gangetic Plain. The TOA forcing follows a similar pattern with slightly lower absolute values. These forcings are associated with regionally and seasonally varying atmospheric cooling and heating, with persistent cooling over large parts of the dust belt from North Africa, the Arabian Peninsula to Pakistan, and heating south of the Sahel, so that a mostly stabilising impact on the atmospheric stratification is expected, which may affect the atmospheric dynamics.

Globally, dust-pollution interactions enhance the net cooling effect of mineral dust on climate. The global, annual average TOA direct radiative forcing of $-0.05$ W/m$^2$ is of similar magnitude as the dust forcing ignoring the interactions, which underscores the importance of a detailed account of these interactions in the assessment of aerosol radiative forcing.

To obtain the direct forcing and to reduce the statistical noise, in the present study we have excluded feedbacks of dust and other aerosol effects on radiation transfer and clouds. These are expected to have a significant impact on atmospheric dynamics and climate, which will be the subject of a subsequent study.

*Code and data availability.* The Modular Earth Submodel System (MESSy) is continuously further developed and applied by a consortium of institutions. The usage of MESSy and access to the source code is licenced to all affiliates of institutions which are members of the MESSy Consortium. Institutions can become a member of the MESSy Consortium by signing the MESSy Memorandum of Understanding. More information can be found on the MESSy Consortium Website (https://www.messy-interface.org). The ECHAM climate model is available to the scientific community under the MPI-M Software License Agreement (https://www.mpimet.mpg.de/en/science/models/license). The simulation results analysed in this study are available at https://edmond.mpdl.mpg.de/imeji/collection/XXXXXXXXXXXXXXXX.

*Author contributions.* KK performed the simulations assisted by VAK, analysed the model output and wrote the manuscript with support from JL who initiated the study. All authors interpreted the results and finalised the manuscript.

*Competing interests.* The authors declare that they have no conflict of interest.

*Acknowledgements.* The research reported in this publication has received funding from the King Abdullah University of Science and Technology (KAUST) CRG3 grant URF/1/2180-01 *Combined Radiative and Air Quality Effects of Anthropogenic Air Pollution and Dust over the Arabian Peninsula*.

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

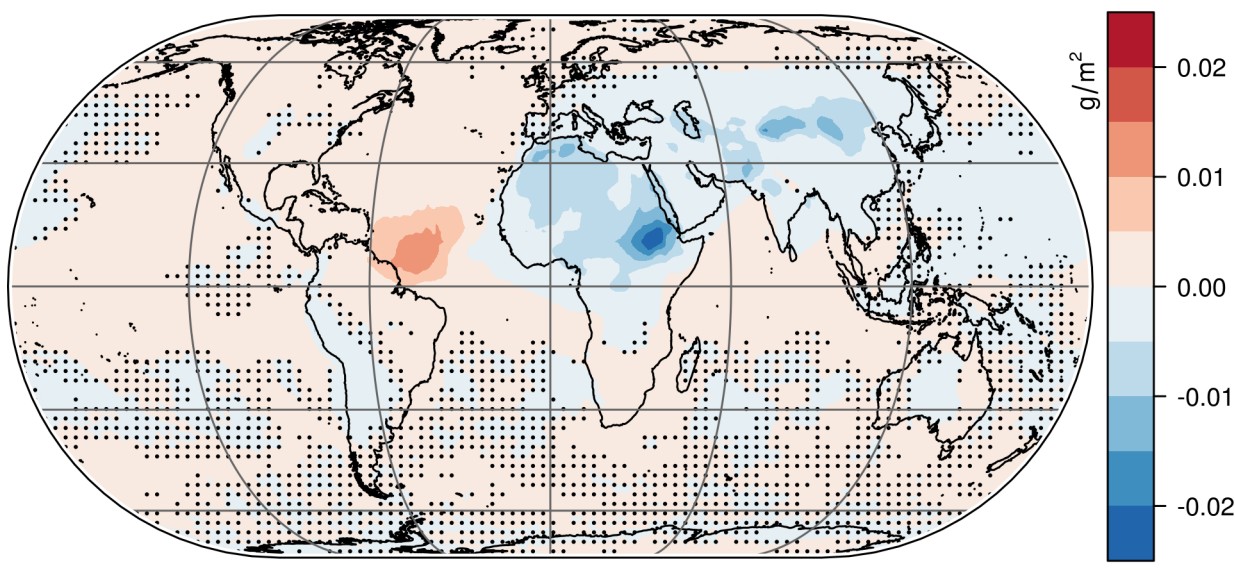

# Δ coarse mode burden

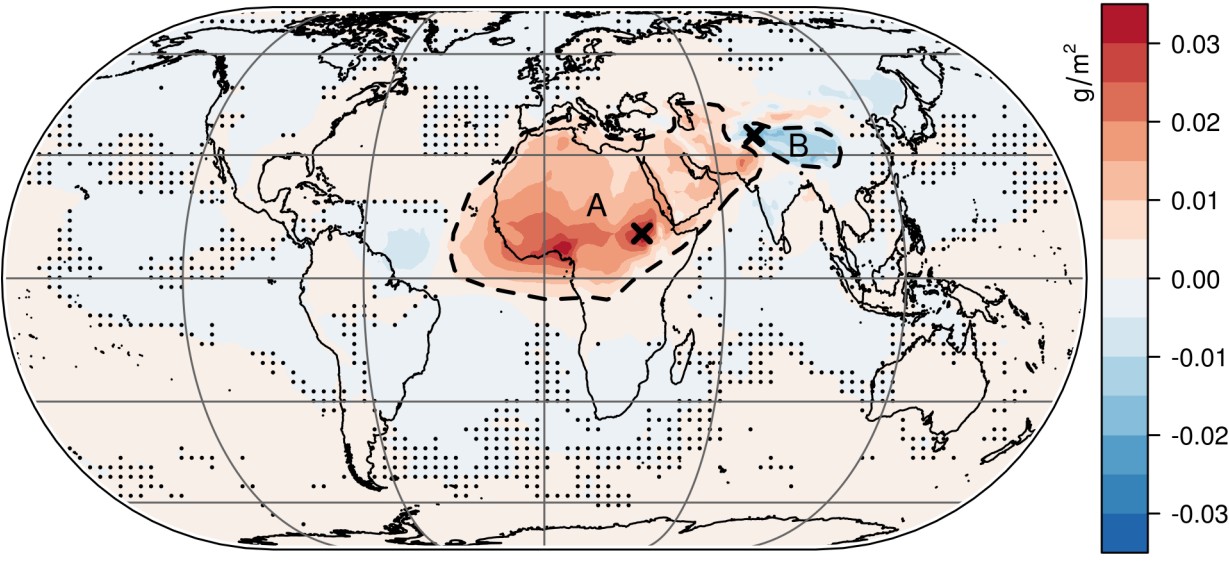

# Δ accumulation mode burden

**Figure 1.** Impact of dust-pollution interaction on the coarse mode (top) and accumulation mode (bottom) aerosol burden (including aerosol water) (based on Eq. (1)). The more efficient removal of aged dust particles reduces the coarse mode burden throughout the dust belt. This in turn reduces the coagulation efficiency of coarse mode with smaller particles, increasing the accumulation mode burden especially where the dust and the African biomass burning regions coincide. The strong hygroscopic growth of aged Saharan dust particles over the western Atlantic results in a regional maximum of the coarse mode burden increase. Generally, the hygroscopic growth of accumulation mode particles is reduced by the interaction with mineral dust cations manifested in a decreased accumulation mode burden over Tibet. The burdens over points A and B (crosses) are analysed in Fig. 2, relevant variables over regions A and B in Table 3 and Fig. S13. Dots indicate regions where the effect of the dust-pollution interaction is insignificant.

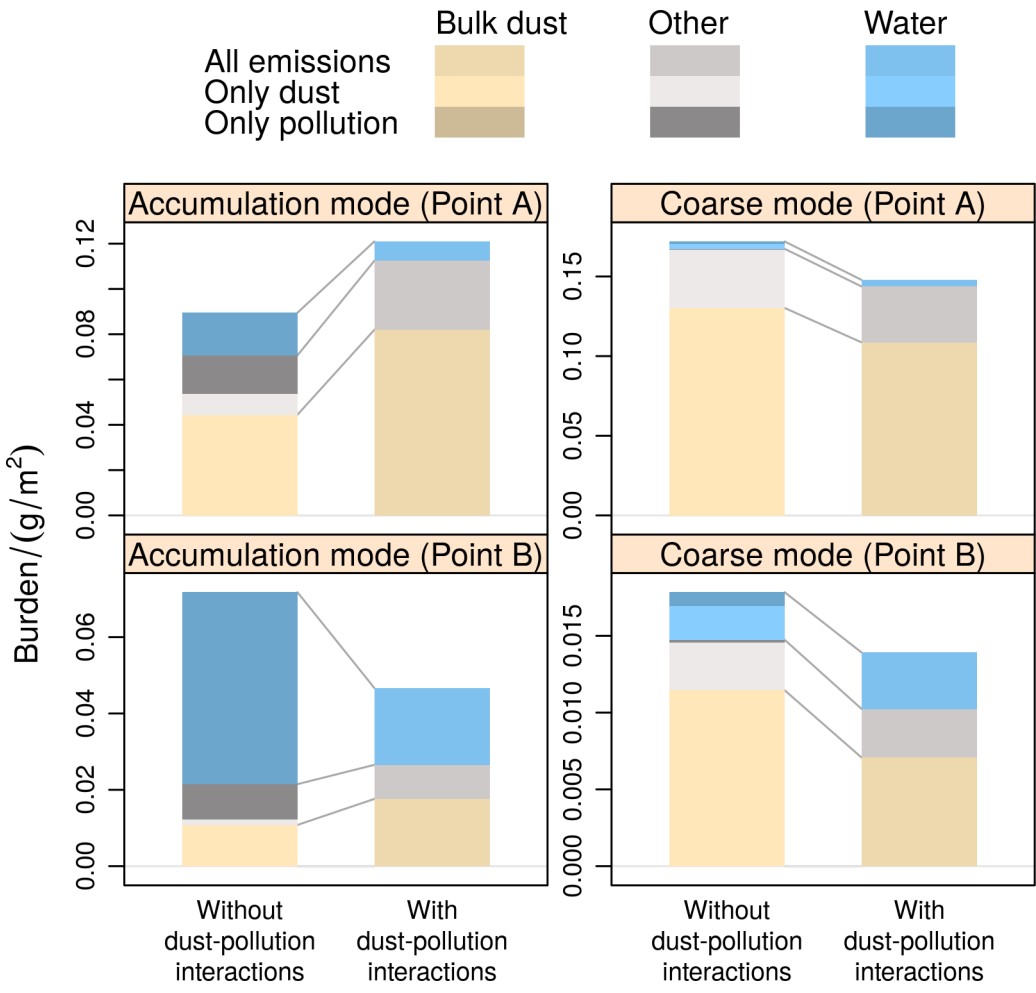

**Figure 2.** Analysis of the interaction effect on the aerosol mass burden over point A (top) and point B (bottom) in Fig. 1. The left bar in each panel represents the burden of non-interacting dust and pollution. It corresponds to $(F_2 - F_4) + (F_3 - F_4)$ in Eq. (2), stacking the burdens in simulation 2 with only pollution and simulation 3 with only dust, each after subtracting the background burdens from simulation 4 without dust and pollution. The right bar in each panel represents the burden of interacting dust and pollution in simulation 1 (after subtracting the background burden) corresponding to $F_1 - F_4$ in Eq. (2).

## ΔAOD (250 to 690 nm)

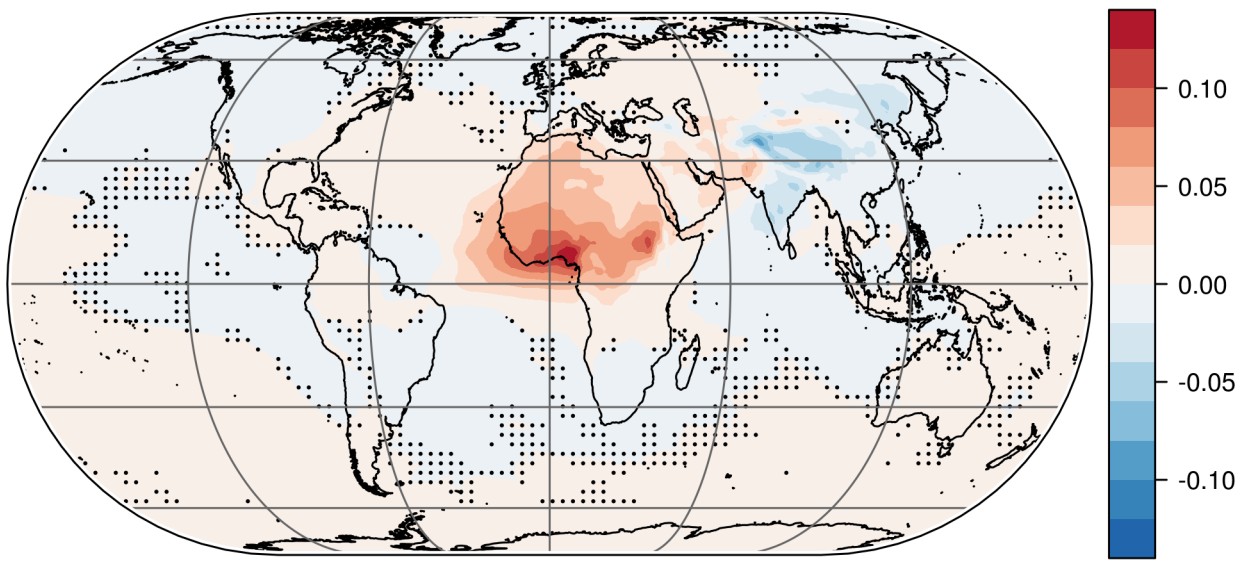

## ΔAAOD (250 to 690 nm)

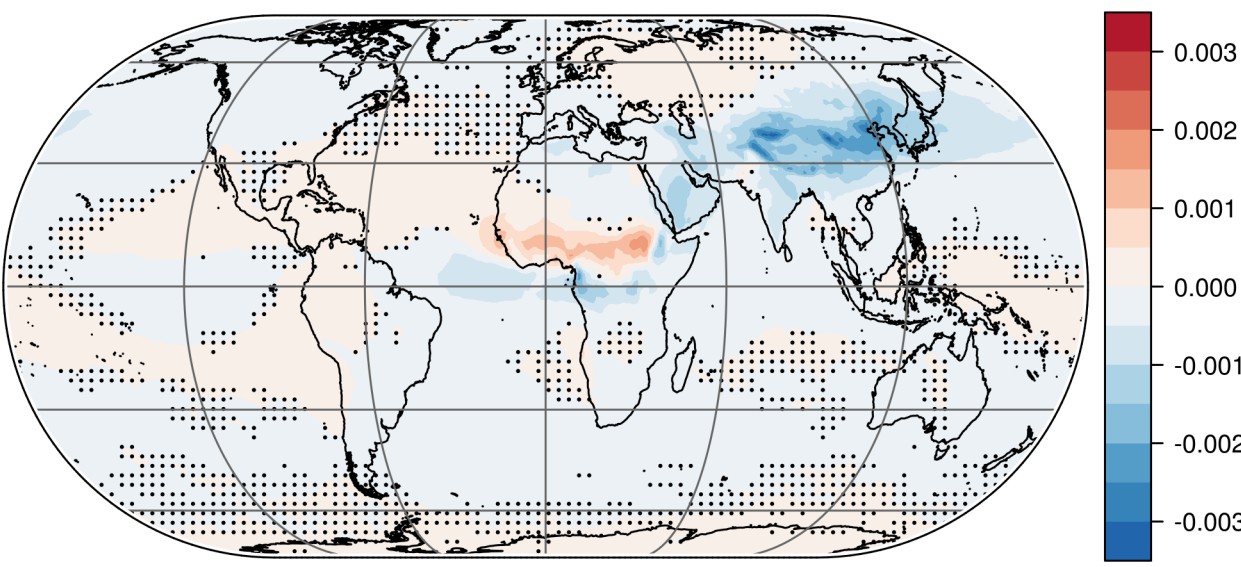

**Figure 3.** Impact of dust-pollution interaction on the AOD (top) and the absorption AOD (AAOD, bottom) (based on Eq. (1)). The AOD change reflects the changes of the accumulation mode burden shown in Fig. 1. Over large parts of the dust belt the accumulation mode AOD is increased. In contrast, over the Tibetan Plateau and east India the AOD decreases due to reduced hygroscopic growth of accumulation mode particles. South of the Sahel, where the Saharan dust mixes with biomass burning pollution, the strongest accumulation mode AOD and AAOD increase occurs. Elsewhere, the water uptake of dust and the more efficient removal of absorbing coarse dust particles combined with the changed accumulation mode particle radii and refractive indices tend to decrease the AAOD. Dots indicate regions where the effect of the dust-pollution interaction is insignificant. Figure S15 in the supplement shows the corresponding plots for the four seasons.

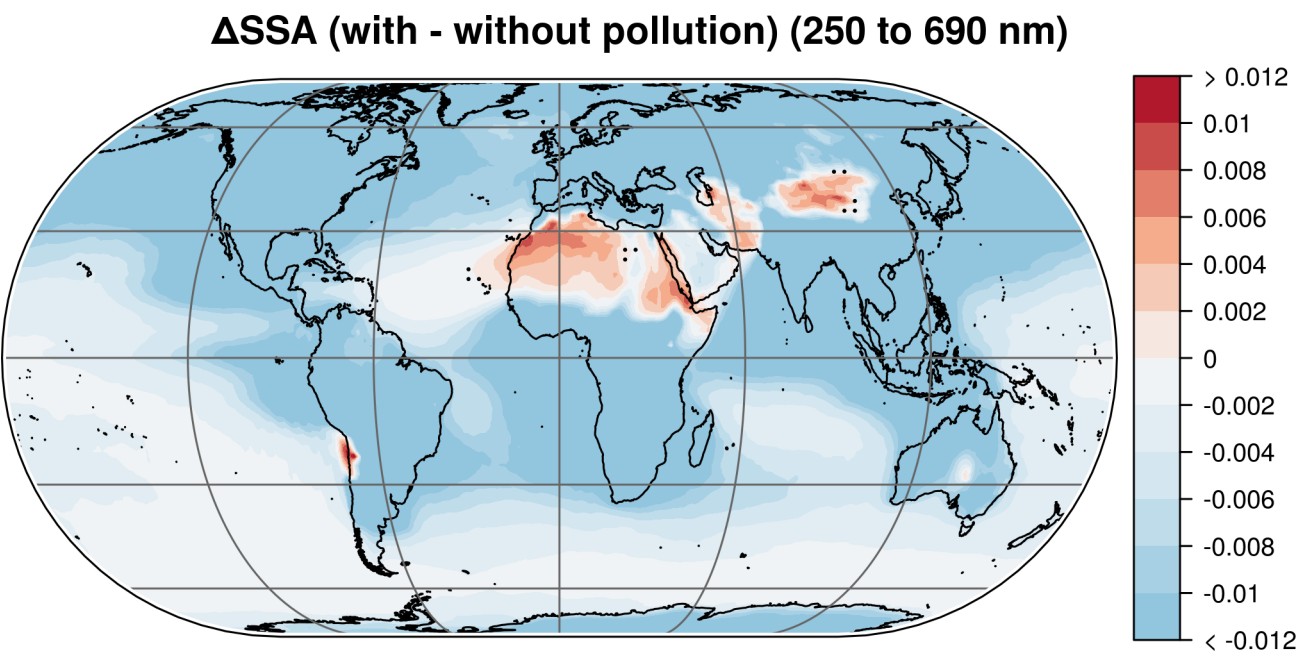

**Figure 4.** Annual mean difference of the single scattering albedo (SSA) with (simulation 1) and without (simulation 3) anthropogenic emissions. Extinction weighted mean SSA values of each vertical column are used. The SSA for all four emission setups is shown in Fig. S16 in the supplement. Dots indicate regions where the difference is insignificant.

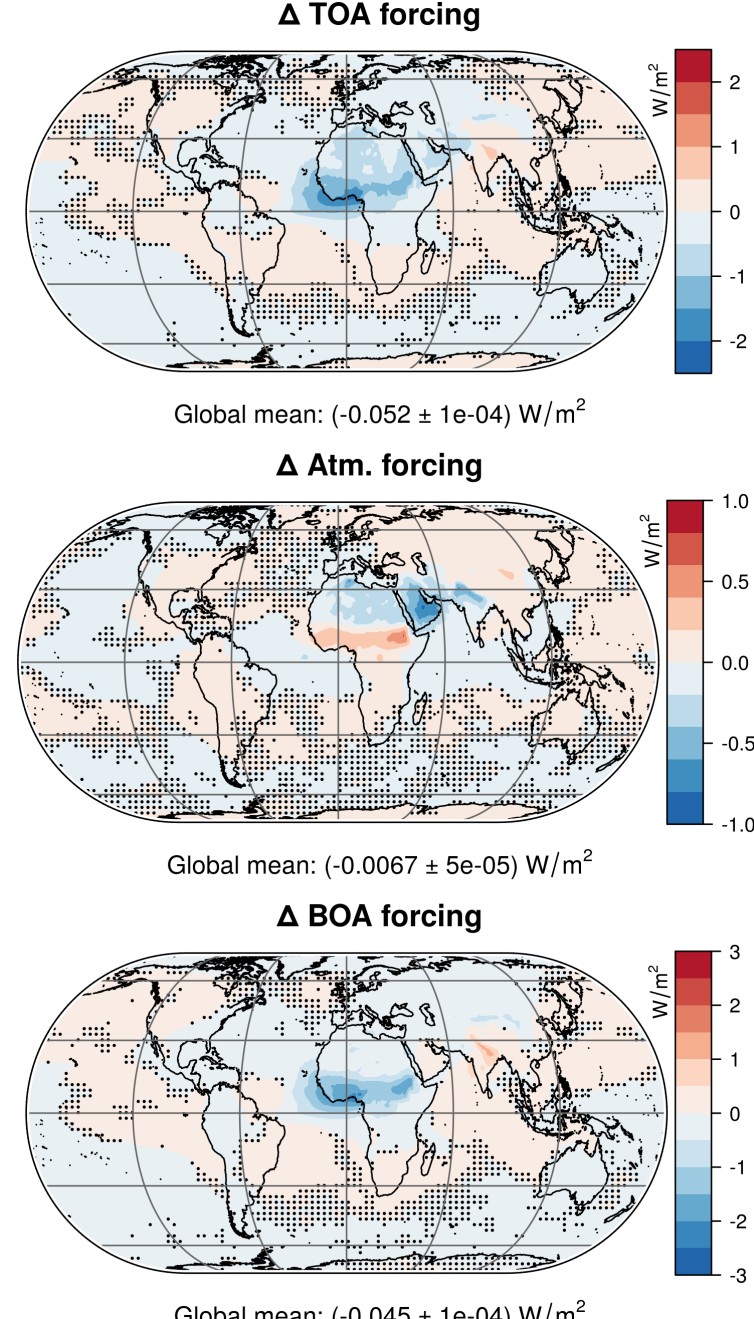

**Figure 5.** The instantaneous total (solar and terrestrial) direct radiative forcing of the dust-pollution interaction at the top of the atmosphere (TOA, top), within the atmosphere (centre) and at the bottom of the atmosphere (BOA, bottom). Dots indicate regions where the effect of the dust-pollution interaction is insignificant. The corresponding figures showing the solar and terrestrial forcings as well as seasonality are provided in the supplement (Figs. S17 to S19).

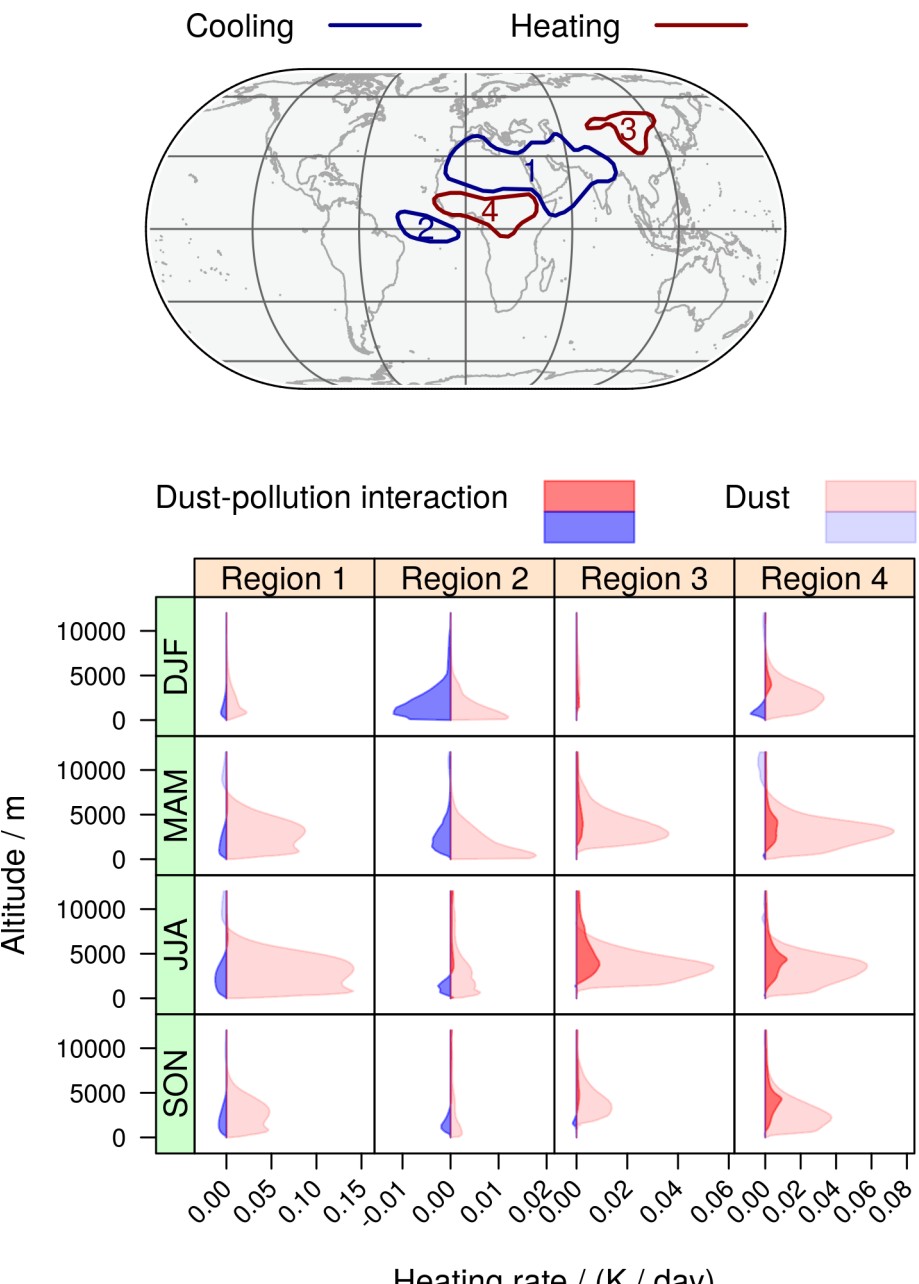

**Figure 6.** Heating rate contribution of dust-pollution interactions (rich colours) in comparison with the mineral dust contribution (pale colours). Seasonal heating rate profiles of four regions with negative (regions 1 and 2) and positive (regions 3 and 4) annual mean atmospheric forcing are shown (bottom). The regions (top) are selected based on the forcings displayed in the centre of Fig. 5, using regions where the absolute forcing exceeds 0.1 W/m$^2$ after applying a Gaussian filter to avoid fragmentation.

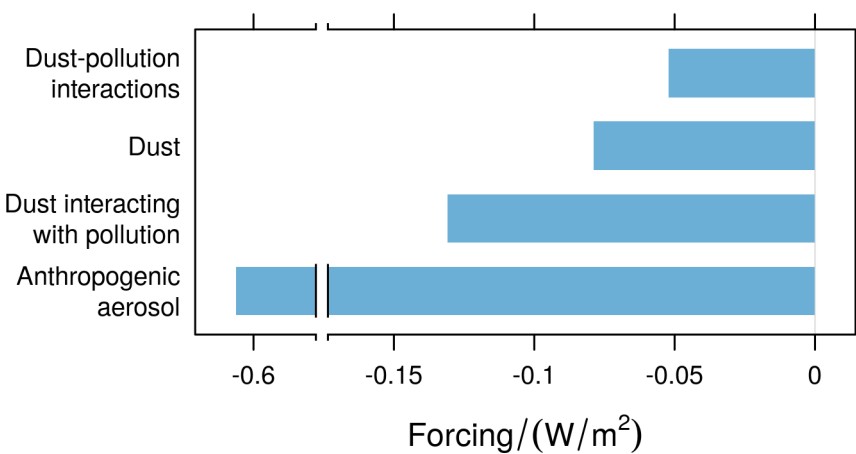

**Figure 7.** Global mean top of the atmosphere (TOA) forcing of the dust-pollution interactions in comparison with the mineral dust forcing from the same EMAC simulation excluding/including the dust-pollution interactions, and the anthropogenic aerosol forcing. The CVs of the interannual variation are (from top to bottom) 7 %, 6 %, 5 % and 2 %.

**Table 1.** Emission setups

| Simulation | 1 | 2 | 3 | 4 |
|---|---|---|---|---|
| Anthropogenic emissions | yes | yes | no | no |
| Dust emissions | yes | no | yes | no |

**Table 2.** Accommodation coefficients

| Gas-phase species | Accommodation coefficient | References |
|---|---|---|
| Sulfuric acid ($H_2SO_4$) | 0.3 (on insoluble particles) | M7 (Vignati et al., 2004), Raes and Van Dingenen (1992) |
| | 1 (on soluble particles) | M7 (Vignati et al., 2004) |
| Nitric acid ($HNO_3$) | 0.1 | GMXE (Pringle et al., 2010a, b), Hanisch and Crowley (2003) |
| Hydrochloric acid (HCl) | 0.064 | GMXE (Pringle et al., 2010a, b), Van Doren et al. (1990) |
| Ammonia ($NH_3$) | 0.097 | GMXE (Pringle et al., 2010a, b), Feng and Penner (2007) |
| Water ($H_2O$) | 0.3 (on insoluble particles) | GMXE (Pringle et al., 2010a, b) |
| | 1 (on soluble particles) | GMXE (Pringle et al., 2010a, b) |

**Table 3.** Annual mean results for various variables over regions A and B in Fig. 1 and the included contributions Δ of the dust-pollution interactions.

| Variable | Region A | | Region B | | Unit |
|---|---|---|---|---|---|
| | Value | Δ | Value | Δ | |
| AOD (250 to 690 nm) | 0.38 | 0.042 | 0.16 | -0.047 | |
| AAOD (250 to 690 nm) | 0.029 | -0.00017 | 0.0094 | -0.0017 | |
| Dust forcing, pollution forcing | -0.35, -1.2 | -0.68 | 0.072, -0.20 | 0.13 | W / m² |
| Mineral dust burden (accumulation mode) | 0.042 | 0.013 | 0.011 | 0.0045 | μg / m³ |
| Mineral dust burden (coarse mode) | 0.096 | -0.0073 | 0.0061 | -0.0024 | μg / m³ |
| BC burden (accumulation mode) | 0.00034 | -8.3e-05 | 0.00021 | -3.1e-05 | μg / m³ |
| BC burden (coarse mode) | 6.8e-05 | 2.3e-05 | 2.9e-05 | 1.2e-05 | μg / m³ |
| Sea salt burden (accumulation mode) | 0.0028 | -0.00064 | 0.00030 | -3.5e-05 | μg / m³ |
| Sea salt burden (coarse mode) | 0.0030 | 0.00019 | 6.6e-05 | -1.6e-05 | μg / m³ |
| Aerosol water burden (accumulation mode) | 0.018 | -0.0011 | 0.023 | -0.015 | μg / m³ |
| Aerosol water burden (coarse mode) | 0.011 | 0.0016 | 0.0040 | -0.00029 | μg / m³ |