# Peer review of "Direct radiative effect of dust-pollution interactions"

_Atmospheric Chemistry and Physics, 2018_

## Referee Comment (RC1) · Anonymous Referee #1 · 11 Dec 2018

This work described how the mixing of dust and pollution aerosol changes dust optical and radiative properties by increasing its extinction and reducing its absorption. Furthermore it quantifies the change in radiative effect of natural dust at the top of the atmosphere, at the surface and in the atmospheric column. A short discussion of regions where the mixed dust cools or warms the atmosphere over 4 regions is also included. To be worthy of being published in ACP, this paper needs to give a reasonable amount of details on how the dust cycle is treated, what are the assumptions under which acids can be uptaken by dust and give the limitation of the optical properties derived by mixing dust with other aerosols. Here are the elements to add to this work to place it in the context of what has already been published on dust.

Ridley et al., 2016 and Kok et al. (2017) that you cite have established constraints

on the dust cycle using both satellite observations and 3 independent models. Please state tha amount of total yearly emissions, the yearly mean total optical depth and total deposition flux of dust in your model and compare them to these constraints.

How do you represent the particle size distribution (PSD) of dust. The PSD has a large influence on the value of your LW radiative effect. You should state how large a particle you represent and what fraction of dust particles suspended are PM1, PM10, or above 10 microns for example.

Then, when you account for the uptake of acids by dust, please state which acids, which chemical species you consider, and what accommodation coefficients were chosen. No need to refer the reader to another paper, a simple Table can go a long way to help the reader navigate through your assumptions of uptake coefficients.

Other major points include:

- Page 2, between the 2 last paragraphs of this page it would be welcome to state the main questions that you are trying to answer in this paper. After reading the introduction, the reader should be fully aware in what directions the paper will take him or her.

Page 3, line 26: The reader needs to know what refractive indices you took for dust both in the SW and in the LW, please state simply your reference for them and whether these refractive indices are coherent with AERONET observations or indicate a dust that is too absorbing compared to these measurements.

Page 3, line 26. You state that you assume spherical particles and make an hypothesis of volume averaged refractive index when mixing particles. There is abundant literature that this approximation in invalid and that a dielectric or a Maxwell-Garnett approximation describes better the state of mixture of an aerosol. State why you made this choice and what is the error associated with it.

Minor points:

In the abstract please state that you treat both dust radiative effect both in the SW and LW range of the spectrum.

A Figure showing as a function of wavelength the refractive indices used separately for the SW and LW would be very useful to the reader.

Page 5 line 20: could you find any field measurements that could guide you as to whether the hygroscopic variations of dust that the model represents are well captured?

Page 7 line 8: typo Change: "The distribution of the bottom of the atmosphere (BOA) forcing (Fig. 4, bottom) is similar to the that of the TOA forcing," with "The distribution of the bottom of the atmosphere (BOA) forcing (Fig. 4, bottom) is similar to that of the TOA forcing,"

Page 7, line 25; I am expecting that for region 1 Figure 5 would show a cooling effect for the atmospheric column. On this Figure, the heating rates are positive at almost all heights, how one can reconcile this fact with an overall cooling effect?

Page 8, lines 11 and 12: please give separately the contributions of the SW and the LW to the pollution free dust radiative effect of -0.08 W .m-2 and to the polluted dust of -0.13 W .m-2.

I believe that this paper can be vastly improved if you account for these recommendations.

References: Kok, J. F., Ridley, D. A., Zhou, Q., Miller, R. L., Zhao, C., Heald, C. L., Ward, D. S., Albani, S., and Haustein, K.: Smaller desert dust cooling effect estimated from analysis of dust size and abundance, Nat. Geosci., 10, 274–278, https://doi.org/10.1038/ngeo2912, 2017.

Ridley, D. A., Heald, C. L., Kok, J. F., and Zhao, C.: An observationally constrained estimate of global dust aerosol optical depth, Atmos. Chem. Phys., 16, 15097-15117, https://doi.org/10.5194/acp-16-15097-2016, 2016.

---

## Referee Comment (RC2) · Anonymous Referee #2 · 20 Dec 2018

General comment

This manuscript investigates how microphysical and chemical interactions between dust and pollution alters the properties of aerosols and their direct radiative effects (DRE). The experimental design is simple and effective. A set of four simulations have been used to model the properties of aerosols when dust and pollution are either emitted separately, or together so that they interact. By contrasting results from these simulations the study reveals the "interaction" term showing how the properties of the dust and the pollution-related aerosol change due to two-way microphysical and chemical interactions. It is interesting to explore such interactions and the changes in aerosol radiative effects are not trivial, so worth noting.

The main result is that the dust-pollution interactions lead to a -0.05 Wm-2 change

in net flux at TOA, dominated by increased SW reflection. This occurs mainly due to increases in accumulation-mode aerosol mass and AOD. The AAOD also drops slightly, though it is not completely clear why, though it is perhaps related to a drop in coarse-mode dust. As climate models and Earth-system models are increasingly adopting more complex microphysical aerosol schemes it is worthwhile understanding what happens as such interactions are enabled. To my knowledge this manuscript is novel and I would judge it to be relevant and worth publishing in ACP. The text is generally well written, well structured and concise. However, significant improvements in the analysis and interpretation of the results are required for the study to be published.

Major comments

The main difficulty with this manuscript is that is it not very clear what has caused the negative change in aerosol DRE. The text interprets this as a change in dust forcing, or an "anthropogenic radiative forcing associated with dust". However, the dust-pollution interaction is a two-way process and changes both the dust and the fine-mode anthropogenic aerosols. Figures S11, S12, S13 indicate significant changes to the "pollution-related" aerosol once the dust and pollution are emitted together so one can not attribute the change in DRE entirely to the dust.

The main cause of the negative change in aerosol DRE seems to be the increases in accumulation-mode aerosol, but it is not very clear from the study which aerosol components have contributed to this increase. More information is required to show how the aerosol properties have changed, including changes in aerosol mass, chemical composition, hygroscopy, and possibly particle size across the relevant size modes.

Related to this, it is not very clear what has caused the changes in AAOD and SSA. Presumably AAOD reduces in dusty regions due to a drop in coarse-mode mass? The AAOD increases across the Sahel are apparently due to increases in BC and OC mass (lifetime) but it would be good to see the evidence to support this. However, it isn't clear why the SSA drops in non-dusty regions that are quite remote from dust sources.

Presumably there is a relative reduction in non-absorbing aerosol components such as sulphate and/or nitrate, but why does this occur in regions very remote from dust. There are some clues in Figure S11 but there are many competing effects and the information is not comprehensive enough to understand what it going on.

The other major concern I have is the short duration of the simulations (only one year). Given the episodic nature of dust emissions there is likely to be considerable inter-annual variability in dust loadings and in how these interact with pollution outbreaks. This could affect both the magnitude and spatial patterns of the results. I would recommend extending the simulations to least a 10 years, unless the authors can provide evidence that a single year is sufficient to gauge the magnitude and characteristics of the dust-pollution interactions.

Minor comments

Abstract: The abstract is short and direct but needs to be altered to reflect the concerns above. In particular, the change in aerosol DRE is described as a "radiative forcing" here and throughout the manuscript. This could be confusing or misleading as the term "forcing" is usually used to indicate a change in radiation balance due to a perturbation in aerosol emissions. The $\Delta F$ is really more "the change in aerosol DRE due to dust-pollution interactions". This could be given a label such as $\Delta DREint$ to avoid using this long definition each time in the text. P1 L5: Please spell out EMAC. P1 L7: Whilst the magnitude of the change in TOA radiation balance is worth noting, I would not describe it as large. In fact it is quite small compared to the total DRE of aerosol in present-day climate ($\sim$ -2 to -3 Wm-2). P1 L10: Please quantify this "considerable fraction".

Methods: P3 L10: It is not quite clear what the term "prognostic radiative-transfer calculations" means. Presumably this is the radiative-transfer calculations that are used to calculate fluxes and heating rates in the simulations. Using "prognostic" is a bit confusing since the prognostic aerosols are clearly not used in the radiation scheme, and a radiation scheme is not itself prognostic. P3 L13: Is the dynamical evolution of

the atmosphere (wind, temperature, moisture, cloud) identical in all four simulations? I would have thought so if the prognostic aerosols neither interact with the clouds or the radiation scheme. P4 L17-18: I think that you need to swap F2 - F4 and F3 − F4 in this sentence. From my reading of the text F3- F4 corresponds to the dust radiative effect and F2 − F4 corresponds to the pollution. P4 L23: So it sounds like a radiation double-call procedure has been used, as outlined in Ghan et al. (2012). https://journals.ametsoc.org/doi/10.1175/JCLI-D-11-00650.1 If so it might be useful to reference Ghan et al. (2012) as this paper outlines the double-call concept fully.

Results: P5 L20 / Fig 1: Does the mass shown in Fig 1 include the aerosol water content? Figures 1 − 3: Exactly how are the differences in mass and optical properties calculated? Are these calculated using the same logic as in equation 1? P5 L24: Would it be possible to provide a figure for the dust mass loading, or at least refer to Figure S1 here so the author can see where the "dust affected regions" are simulations. Section 3: I found this section difficult to follow (particularly the top half of P6) and it did not provide a full explanation of how / why the aerosol properties changed.

P6 L7-9: This argument needs explaining more fully. It is clear that the the coarse-mode dust is removed more rapidly due to secondary aerosol forming on the particles leading to more rapid wet deposition. However, what happens to the accumulation-mode dust? Wouldn't the same process also speed the removal of the accumulation-mode dust compared to the simulation where pollution was not emitted with the dust? From figure S11 it looks like the overall mass of sulphate and nitrate aerosol in the accumulation-mode has decreased. So this would tend to decrease accumulation-mode mass and the hygroscopicity, yet total accumulation-mode mass and AOD have increased. Presumably the mass of BC and OC must have increased dramatically to compensate the decreases in sulphate and nitrate. Is there evidence of this? It would be good to see all the relevant mass components in figure S11 and have the full story explained.

P6 L25-26: The SSA has reduced in most non-dusty regions. Is this due to an increase

in the relative proportion of BC and/or OC versus sulphate and nitrate? Why would this have occurred even in regions very remote from dust? It would be good to provide a table listing the global-mean values and global mean changes in relevant quantities, such as AOD, AAOD, radiative flux changes and the various aerosol mass components.

P6 L20: How has the aerosol become more reflective? I suspect the drop in AAOD is dominated by the decline in coarse-mode dust mass so I would omit "due to the higher reflectance" from this sentence. P6 L21. The AAOD is presumably increased in the Sahel due to increased BC mass (and brown carbon if this is included in the model). The explanation that this is due to AOD increase doesn't make sense, only one can say that both may be increasing for similar reasons.

Conclusions: The main concerns given above need to be addressed throughout the conclusions. The change in aerosol DRE can not be interpreted as a change in dust forcing as it is caused by both changes in dust and pollution aerosol properties. The conclusions section is very concise, which is good, but a bit more discussion is required to explain what has caused the changes in aerosol DRE, and how these are linked to aerosol processes and changes in aerosol properties (mass, hygroscopicity, optical properties etc). P8 L20: The maximum impact to the south of the Sahel is given as -2.5 Wm-2 here but -2Wm-2 in the abstract. From reading closer I see this is because the -2.5Wm-2 quoted here corresponds to the surface forcing and the abstract gives the TOA forcing. Please use the same headline result in abstract and conclusions.

Figures: There are a lot of additional figures in the supplementary material and some offer an unnecessary level of detail on the spatial and seasonal variability of aerosol radiative effects (S5, S9, S15, S19 - 23). Given that the spatial distributions and seasonality of results are probably very specific to the model and the meteorological evolution in this specific set of simulations, this level of detail is not particularly useful and could be misleading. The 3D visualizations of aerosol heating in particular are not at all useful.

Figure S11: As expressed above, to really understand how the dust-pollution interaction affects aerosol properties this figure needs to include changes in OC, BC, dust, water and sea salt. The figure may need expanding to two or three figures to give a complete summary of the changes in aerosol mass and composition.

Figure S12 & S13: It is quite interesting to have this kind of information, but it was rather difficult to interpret how the mass of dust, pollution, water and natural aerosols change between the three scenarios. It would be clearer to have just one bar for each emission scenario (all emission, no pollution, no dust) and have each bar stacked showing the relevant mass components (dust, pollution, water and other natural aerosol). This way it would be totally clear how each mass contribution has changed depending on what has been emitted. It is still very interesting to provide this separately for both the accumulation and coarse-modes. Would it be possible to produce this kind of figure also for the global mean changes? As this analysis is really quite important to the story of the paper it would also be worth considering moving this (especially a plot with global-mean changes) to the main article.

Figure 6: This graph really emphasizes the interpretation that the dust-pollution interaction has strengthened dust forcing, which is misleading since the interaction has also altered the strength of the anthropogenic (pollution) aerosol forcing. For a more balanced summary it would be good to include a bar for the forcing from pollution and bars for "Dust + pollution with interaction", and "Dust + pollution without interaction" instead of the bar with "Dust interacting with pollution".

---

## Author Comment (AC1) · 27 Mar 2019

**Reply to referee 1**

We thank the referee for the comprehensive and constructive review which helped a lot to revise the manuscript. Below please find our point-by-point reply to the comments which are repeated here in shaded boxes.

> This work described how the mixing of dust and pollution aerosol changes dust optical and radiative properties by increasing its extinction and reducing its absorption. Furthermore it quantifies the change in radiative effect of natural dust at the top of the atmosphere, at the surface and in the atmospheric column. A short discussion of regions where the mixed dust cools or warms the atmosphere over 4 regions is also included. To be worthy of being published in ACP, this paper needs to give a reasonable amount of details on how the dust cycle is treated, what are the assumptions under which acids can be uptaken by dust and give the limitation of the optical properties derived by mixing dust with other aerosols. Here are the elements to add to this work to place it in the context of what has already been published on dust.

We have included more details on the implementation of the dust cycle, the condensation of acids and the aerosol optical properties in section 2 of the revised manuscript.

> Ridley et al., 2016 and Kok et al. (2017) that you cite have established constraints on the dust cycle using both satellite observations and 3 independent models. Please state tha amount of total yearly emissions, the yearly mean total optical depth and total deposition flux of dust in your model and compare them to these constraints.

In the revised manuscript we discuss these numbers in the last paragraph of the methodology section.

> How do you represent the particle size distribution (PSD) of dust. The PSD has a large influence on the value of your LW radiative effect. You should state how large a particle you represent and what fraction of dust particles suspended are PM1, PM10, or above 10 microns for example.

We have added information on the modal size distribution to the methodology section. One indicator for a realistic size distribution are the comparisons with AOD observations at different wavelengths shown in Fig. S1 in the supplement.

> Then, when you account for the uptake of acids by dust, please state which acids, which chemical species you consider, and what accommodation coefficients were chosen. No need to refer the reader to another paper, a simple Table can go a long way to help the reader navigate through your assumptions of uptake coefficients.

We have added Table 2.

> Other major points include:
> Page 2, between the 2 last paragraphs of this page it would be welcome to state the main questions that you are trying to answer in this paper. After reading the introduction, the reader should be fully aware in what directions the paper will take him or her.

We inserted a new paragraph.

> Page 3, line 26: The reader needs to know what refractive indices you took for dust both in the SW and in the LW, please state simply your reference for them and whether these refractive indices are coherent with AERONET observations or indicate a dust that is too absorbing compared to these measurements.

We have added the sources of all refractive indices used by the AEROPT submodel and added the refractive index figure and tables from the supplement of Klingmueller et al. (2014) to the supplement.

> Page 3, line 26. You state that you assume spherical particles and make an hypothesis of volume averaged refractive index when mixing particles. There is abundant literature that this approximation in invalid and that a dielectric or a Maxwell-Garnett approximation describes better the state of mixture of an aerosol. State why you made this choice and what is the error associated with it.

Using the volume average refractive index is the EMAC default mixing rule and has been applied in previous studies on which the present study is based. More importantly, comparing various mixing assumptions in EMAC (Klingmueller et al., 2014), where the optical properties integrate a large range of Mie size parameters and particle compositions, showed that using the Maxwell-Garnett mixing rule does not much change the results. In addition, in the same study the average refractive index mixing rule tends to yield optical properties closer to that of core-shell particles than the Maxwell-Garnett mixing rule. Considering that under certain conditions core shell particles represent real particles (in particular dust after the uptake of water) more closely than homogeneous mixed particles, there is no clear advantage in using the Maxwell-Garnett mixing rule for our application.

> Minor points:
> In the abstract please state that you treat both dust radiative effect both in the SW and LW range of the spectrum.

This is mentioned in the revised abstract.

> A Figure showing as a function of wavelength the refractive indices used separately for the SW and LW would be very useful to the reader.

We have added the refractive index figure and tables from the supplement of Klingmueller et al. (2014) to the supplement.

> Page 5 line 20: could you find any field measurements that could guide you as to whether the hygroscopic variations of dust that the model represents are well captured?

The model representation of the hygroscopic variations has been evaluated against field measurements by Metzger et al. 2016. Abdelkader 2017 evaluated the chemical ageing during the transatlantic dust transport using ground based AERONET observations and satellite retrievals from MODIS and CALIPSO. Consistent results where also reported by other studies (e.g., Abdelkader 2015, Klingmueller et al. 2018, Bruehl et al. 2018).

> Page 7 line 8: typo Change: "The distribution of the bottom of the atmosphere (BOA) forcing (Fig. 4, bottom) is similar to the that of the TOA forcing," with "The distribution of the bottom of the atmosphere (BOA) forcing (Fig. 4, bottom) is similar to that of the TOA forcing,"

This has been fixed in the revision.

> Page 7, line 25; I am expecting that for region 1 Figure 5 would show a cooling effect for the atmospheric column. On this Figure, the heating rates are positive at almost all heights, how one can reconcile this fact with an overall cooling effect?

The profiles depicted in pale colours are the heating rates caused by dust which is now explicitly mentioned in the caption. The heating rates due to the interactions are negative (blue).

> Page 8, lines 11 and 12: please give separately the contributions of the SW and the LW to the pollution free dust radiative effect of -0.08 W .m-2 and to the polluted dust of -0.13 W .m-2.

We have added the SW and LW contributions.

> I believe that this paper can be vastly improved if you account for these recommendations.
> References: Kok, J. F., Ridley, D. A., Zhou, Q., Miller, R. L., Zhao, C., Heald, C. L., Ward, D. S., Albani, S., and Haustein, K.: Smaller desert dust cooling effect estimated from analysis of dust size and abundance, Nat. Geosci., 10, 274–278, https://doi.org/10.1038/ngeo2912, 2017. Printer-friendly version
> Ridley, D. A., Heald, C. L., Kok, J. F., and Zhao, C.: An observationally constrained estimate of global dust aerosol optical depth, Atmos. Chem. Phys., 16, 15097-15117, https://doi.org/10.5194/acp-16-15097-2016, 2016.

We use both references in the revised manuscript.

---

## Author Comment (AC2) · 27 Mar 2019

**Reply to referee 2**

We thank the referee for the thorough and constructive review. We have revised the manuscript accordingly. Please find below the point by point reply to the individual comments.

> General comment
> This manuscript investigates how microphysical and chemical interactions between dust and pollution alters the properties of aerosols and their direct radiative effects (DRE). The experimental design is simple and effective. A set of four simulations have been used to model the properties of aerosols when dust and pollution are either emitted separately, or together so that they interact. By contrasting results from these simulations the study reveals the "interaction" term showing how the properties of the dust and the pollution-related aerosol change due to two-way microphysical and chemical interactions. It is interesting to explore such interactions and the changes in aerosol radiative effects are not trivial, so worth noting.
> The main result is that the dust-pollution interactions lead to a -0.05 Wm-2 change in net flux at TOA, dominated by increased SW reflection. This occurs mainly due to increases in accumulation-mode aerosol mass and AOD. The AAOD also drops slightly, though it is not completely clear why, though it is perhaps related to a drop in coarse-mode dust. As climate models and Earth-system models are increasingly adopting more complex microphysical aerosol schemes it is worthwhile understanding what happens as such interactions are enabled. To my knowledge this manuscript is novel and I would judge it to be relevant and worth publishing in ACP. The text is generally well written, well structured and concise. However, significant improvements in the analysis and interpretation of the results are required for the study to be published.
> Major comments
> The main difficulty with this manuscript is that is it not very clear what has caused the negative change in aerosol DRE. The text interprets this as a change in dust forcing, or an "anthropogenic radiative forcing associated with dust". However, the dust-pollution interaction is a two-way process and changes both the dust and the fine-mode anthropogenic aerosols. Figures S11, S12, S13 indicate significant changes to the "pollution-related" aerosol once the dust and pollution are emitted together so one can not attribute the change in DRE entirely to the dust.

Indeed the interaction is a two-way process and technically we treat both directions equally as manifested by Eq. (2). The interpretation as a change in dust forcing is motivated by the fact that historically the dust was there before the pollution and therefore the pollution modified the already existing dust forcing. In contrast, the opposite case where dust is added to pollution never occurred on global scale. Therefore and to make the discussion of the results more comprehensible, we use this interpretation as guiding principle but nevertheless discuss the impact of dust on pollution. The term "anthropogenic radiative forcing associated with dust" is supposed to reflect that the effect is linked to (but not solely attributed to) dust and at the same time anthropogenic because of the anthropogenic origin of the pollution.

> The main cause of the negative change in aerosol DRE seems to be the increases in accumulation-mode aerosol, but it is not very clear from the study which aerosol components have contributed to this increase. More information is required to show how the aerosol properties have changed, including changes in aerosol mass, chemical composition, hygroscopity, and possibly particle size across the relevant size modes.

The increase of the accumulation-mode burden is caused by mineral dust due to the reduced coagulation with coarse particles in the presence of pollution. The accumulation mode-burden of other components is generally reduced due to the increase of coagulation in the presence of dust, exceptions are the ions (Fig. S11, now S12). The increase of accumulation mode dust dominates in dusty regions, resulting in a net-increase of the accumulation-mode burden. We have added figures showing the interaction effect on dust, BC, SS and water burdens separately to the supplement and discuss them in the main text.

> Related to this, it is not very clear what has caused the changes in AAOD and SSA. Presumably AAOD reduces in dusty regions due to a drop in coarse-mode mass? The AAOD increases across the Sahel are apparently due to increases in BC and OC mass (lifetime) but it would be good to see the evidence to support this. However, it isn't clear why the SSA drops in non-dusty regions that are quite remote from dust sources. Presumably there is a relative reduction in non-absorbing aerosol components such as sulphate and/or nitrate, but why does this occur in regions very remote from dust. There are some clues in Figure S11 but there are many competing effects and the information is not comprehensive enough to understand what it going on.

The AAOD reduction in dusty regions is mainly caused by the change of the accumulation mode composition in the presence of dust which transfers absorbing carbonaceous components from the accumulation to the coarse mode. This becomes more clear from the new aerosol component burden plots in the supplement. On the other hand the increased accumulation mode dust increases the AAOD, this effect dominates south of the Sahel to produce a net AAOD increase. In contrast, over Asia where not only the burden of absorbing components but also the AOD decreases, both effects reduce the AAOD so that a negative net effect is obtained even relatively remote from dust sources. Additionally, as you mention, in dusty regions the coarse mode mineral dust and hence an absorbing component is reduced, which further contributes to the AAOD decrease.

> The other major concern I have is the short duration of the simulations (only one year). Given the episodic nature of dust emissions there is likely to be considerable interannual variability in dust loadings and in how these interact with pollution outbreaks. This could affect both the magnitude and spatial patterns of the results. I would recommend extending the simulations to least a 10 years, unless the authors can provide evidence that a single year is sufficient to gauge the magnitude and characteristics of the dust-pollution interactions.

Despite the episodic nature of dust emissions the global annual averages presented here are relatively robust regarding interannual variations which is confirmed by a lower resolution simulation over 10 years that yields coefficients of variation (CV) for our main results below 10 %. Therefore, even though the results are explicitly presented for 2011, they are considered representative for recent years. We have added a paragraph to the methodology section and CV estimates to the caption of Fig. 6.

> Minor comments
> Abstract: The abstract is short and direct but needs to be altered to reflect the concerns above. In particular, the change in aerosol DRE is described as a "radiative forcing" here and throughout the manuscript. This could be confusing or misleading as the term "forcing" is usually used to indicate a change in radiation balance due to a perturbation in aerosol emissions. The $\Delta F$ is really more "the change in aerosol DRE due to dust-pollution interactions". This could be given a label such as $\Delta DRE_{int}$ to avoid using this long definition each time in the text.

While a "forcing" is often attributed to emissions, it is not uncommon to assign a forcing to other effects such as changes in the solar irradiance, surface albedo changes or cloud adjustments due to aerosols. Especially the latter, the forcing of aerosol-cloud interactions, bears similarities to the forcing by dust-pollution interactions so that we consider using the term "forcing" in our context to be not exceptionally confusing.

> P1 L5: Please spell out EMAC.

We have expanded the acronym.

> P1 L7: Whilst the magnitude of the change in TOA radiation balance is worth noting, I would not describe it as large. In fact it is quite small compared to the total DRE of aerosol in present-day climate (~ -2 to -3 Wm-2).

We have deleted "large".

> P1 L10: Please quantify this "considerable fraction".

We have quantified the fraction (40 %).

> Methods: P3 L10: It is not quite clear what the term "prognostic radiative-transfer calculations" means. Presumably this is the radiative-transfer calculations that are used to calculate fluxes and heating rates in the simulations. Using "prognostic" is a bit confusing since the prognostic aerosols are clearly not used in the radiation scheme, and a radiation scheme is not itself prognostic.

We have reformulated to "the aerosol radiative effect on the dynamics is computed using the extinction, single scattering albedo and asymmetry factor from the Tanre aerosol climatology".

> P3 L13: Is the dynamical evolution of the atmosphere (wind, temperature, moisture, cloud) identical in all four simulations? I would have thought so if the prognostic aerosols neither interact with the clouds or the radiation scheme.

Yes, they are identical which is now mentioned in the discussion of Eqs. (1) and (2) in section 2.

> P4 L17-18: I think that you need to swap F2 - F4 and F3 – F4 in this sentence. From my reading of the text F3- F4 corresponds to the dust radiative effect and F2 – F4 corresponds to the pollution.

This has been fixed in the revision.

> P4 L23: So it sounds like a radiation double-call procedure has been used, as outlined in Ghan et al. (2012). https://journals.ametsoc.org/doi/10.1175/JCLI-D-11-00650.1 If so it might be useful to reference Ghan et al. (2012) as this paper outlines the double-call concept fully.

We have added the reference.

> Results: P5 L20 / Fig 1: Does the mass shown in Fig 1 include the aerosol water content?

Yes (we have added a note in the caption of Fig. 1).

> Figures 1 – 3: Exactly how are the differences in mass and optical properties calculated? Are these calculated using the same logic as in equation 1?

Figures 1 and 2 are calculated using Eq. (1). Fig. 3 shows the difference of the SSA results from the simulation with (simulation 1) and without pollution (simulation 3) (both simulations include dust). Equation (1) cannot be used to study the effect on the SSA since the SSAs of dust and pollution are not additive when neglecting the interaction.

> P5 L24: Would it be possible to provide a figure for the dust mass loading, or at least refer to Figure S1 here so the author can see where the "dust affected regions" are simulations.

We have added the reference to Fig. S1.

Section 3: I found this section difficult to follow (particularly the top half of P6) and it did not provide a full explanation of how / why the aerosol properties changed.

P6 L7-9: This argument needs explaining more fully. It is clear that the the coarse-mode dust is removed more rapidly due to secondary aerosol forming on the particles leading to more rapid wet deposition. However, what happens to the accumulation-mode dust? Wouldn't the same process also speed the removal of the accumulation-mode dust compared to the simulation where pollution was not emitted with the dust? From figure S11 it looks like the overall mass of sulphate and nitrate aerosol in the accumulation-mode has decreased. So this would tend to decrease accumulation-mode mass and the hygroscopicity, yet total accumulation-mode mass and AOD have increased. Presumably the mass of BC and OC must have increased dramatically to compensate the decreases in sulphate and nitrate. Is there evidence of this? It would be good to see all the relevant mass components in figure S11 and have the full story explained.

Compared to the coarse mode, an additional sink for the accumulation mode is the coagulation with coarse particles, which is less efficient in the polluted simulation 1 than in simulation 3 without pollution (due to the more efficient removal of coarse particles). Therefore simulation 1 produces more accumulation mode dust particles than simulation 3 despite a more efficient deposition. This difference dominates Eq. (1) resulting in the burden increase displayed in Fig. (1). We have added the corresponding plots for mineral dust, black carbon, sea salt and water individually in Fig. S12 in the supplement, showing that the accumulation mode burden increase is due to dust.

P6 L25-26: The SSA has reduced in most non-dusty regions. Is this due to an increase in the relative proportion of BC and/or OC versus sulphate and nitrate? Why would this have occurred even in regions very remote from dust? It would be good to provide a table listing the global-mean values and global mean changes in relevant quantities, such as AOD, AAOD, radiative flux changes and the various aerosol mass components.

Please note that Fig. 3 does not show the result of Eq. (1) but the SSA difference between simulation 1 with pollution and simulation 3 without pollution. The negative values very remote from dust result from adding pollution including absorbing BC and OC to the mostly non-absorbing natural background aerosol (e.g., sea salt, water). Since the natural aerosol burden can be very low compared to the burden from pollution, the SSA difference in remote regions is not very relevant (hence the colour scale cut-off at -0.012). Because global means integrate compensating effects and large areas unaffected by dust, regional values are more informative (the relevance of the global TOA forcing for the climate's energy budged make it an exception), accordingly we have added a table with regional mean values and the corresponding contributions of the dust-pollution interactions for relevant quantities.

P6 L20: How has the aerosol become more reflective? I suspect the drop in AAOD is dominated by the decline in coarse-mode dust mass so I would omit "due to the higher reflectance" from this sentence.

The water which is taken up by the aged hygroscopic dust particles reduces the average imaginary

refractive index of the particles, increasing the reflectance.

> P6 L21. The AAOD is presumably increased in the Sahel due to increased BC mass (and brown carbon if this is included in the model). The explanation that this is due to AOD increase doesn't make sense, only one can say that both may be increasing for similar reasons.

The AAOD is proportional to the AOD (for constant SSA), thus an increase of the AOD increases the AAOD.

> Conclusions: The main concerns given above need to be addressed throughout the conclusions. The change in aerosol DRE can not be interpreted as a change in dust forcing as it is caused by both changes in dust and pollution aerosol properties. The conclusions section is very concise, which is good, but a bit more discussion is required to explain what has caused the changes in aerosol DRE, and how these are linked to aerosol processes and changes in aerosol properties (mass, hygroscopicity, optical properties etc).

While we agree that the impact of dust on pollution is crucial which is now emphasized more throughout the revised manuscript, the change in the forcing is by definition the change of the dust forcing which occurs when pollution is added to the natural scenario (see Eq. (1)). Mathematically this is identical to the change of the pollution forcing when adding dust to a dust free scenario (($F_1$ - $F_3$) - ($F_2$ - $F_4$)). However, as mentioned above, this never occurred in the real world on a global scale, therefore we consider the interpretation as change in dust forcing to be more instructive. We have added a paragraph summarising the causes of the changes in aerosol forcing.

> P8 L20: The maximum impact to the south of the Sahel is given as -2.5 Wm-2 here but -2 Wm-2 in the abstract. From reading closer I see this is because the -2.5 Wm-2 quoted here corresponds to the surface forcing and the abstract gives the TOA forcing. Please use the same headline result in abstract and conclusions.

We have included the TOA result in the revised conclusions.

> Figures: There are a lot of additional figures in the supplementary material and some offer an unnecessary level of detail on the spatial and seasonal variability of aerosol radiative effects (S5, S9, S15, S19 - 23). Given that the spatial distributions and seasonality of results are probably very specific to the model and the meteorological evolution in this specific set of simulations, this level of detail is not particularly useful and could be misleading. The 3D visualizations of aerosol heating in particular are not at all useful.

It is true that due to the shorter underlying time period the seasonal results have a higher statistical uncertainty and are less representative for the same season of other years. But since the meteorology is nudged towards reanalysis data, the meteorological evolution in the simulations agrees well with observations and so do the aerosol concentrations which is supported by many previous studies (e.g., Pozzer et al. 2012, Abdelkader et al. 2015, 2017, Metzger et al. 2016, Klingmueller et al. 2018). Therefore we consider the seasonal results to be not specific to the simulations but well representative for the year 2011. Given the seasonality of dust emissions we assume the seasonal results are valuable to some readers and prefer to retain most of the figures while removing the 3D plots.

> Figure S11: As expressed above, to really understand how the dust-pollution interaction affects aerosol properties this figure needs to include changes in OC, BC, dust, water and sea salt. The figure may need expanding to two or three figures to give a complete summary of the changes in aerosol mass and composition.

Figure S11 focusses on the main aerosol ions which interact with the mineral cations in the dust and are in exchange with precursor gasses shown in the same figure and therefore play a distinct role. The revised supplement additionally includes the Figures for dust, BC, SS and water burdens separately (Fig. S11) and a discussion in the main text. Qualitatively the same effect as for BC is obtained for OC but no OC coarse mode model output is available.

> Figure S12 & S13: It is quite interesting to have this kind of information, but it was rather difficult to interpret how the mass of dust, pollution, water and natural aerosols change between the three scenarios. It would be clearer to have just one bar for each emission scenario (all emission, no pollution, no dust) and have each bar stacked showing the relevant mass components (dust, pollution, water and other natural aerosol). This way it would be totally clear how each mass contribution has changed depending on what has been emitted. It is still very interesting to provide this separately for both the accumulation and coarse-modes. Would it be possible to produce this kind of figure also for the global mean changes? As this analysis is really quite important to the story of the paper it would also be worth considering moving this (especially a plot with global-mean changes) to the main article.

Figures S12 and S13 are closely related to Eq. (2), the "No dust" and "No pollutions" bars are stacked to allow a direct comparison of the term corresponding to (F2 - F4) + (F3 - F4) with the term corresponding to F1 - F4. We have straightened the figure by stacking the mass contribution bars, but still stacking the "No dust" and "No pollution" values and moved it to the main article. This is complemented by a new figure in the supplement showing regional burdens (which are more useful than global averages) with one bar per simulation as proposed.

> Figure 6: This graph really emphasizes the interpretation that the dust-pollution interaction has strengthened dust forcing, which is misleading since the interaction has also altered the strength of the anthropogenic (pollution) aerosol forcing. For a more balanced summary it would be good to include a bar for the forcing from pollution and bars for "Dust + pollution with interaction", and "Dust + pollution without interaction" instead of the bar with "Dust interacting with pollution".

As argued above, while we agree that the interpretation as a change in dust forcing is not the only interpretation possible, we consider it to be a valid and well motivated perspective. Aside from this the comparison with the dust forcing works better due the similar magnitude. The underlying reason is that for both, the interaction and the dust forcing, regional contributions

compensate each other. The pollution forcing is discussed in section 2.

---

## Author Response (AR2)

Dear Editor

We have addressed the issues raised in the latest review as indicated in the following point by point reply.

Additionally we have included a place holder for the final link to the data repository in the "Code and data availability" section, the final address will be available shortly.

The changes to the manuscript are highlighted in the latexdiff output at the end of this document.

Kind regards

Klaus Klingmueller

**Reply to referee 2**

> The authors have made substantial improvement to this manuscript. The majority of comments have been well addressed making the article clearer and more thoroughly supported. A number of new figures (Fig 2, S11, S13) and sections of discussion have really helped to explain how and why the aerosols change when pollution-dust interactions are included. In particular it is now much clearer that the global-mean negative forcing is dominated by an increase in accumulation mode dust that wins out over reductions in carbonaceous aerosol and coarse-mode dust. The description of experimental methods has also been improved addressing a number of minor comments made in the reviews.

> The only sticking point I have is that the text still refer to the dust-pollution interaction an "anthropogenic forcing associated with mineral dust". It is clear throughout the manuscript that the dust-pollution interaction is a two way process and that both aerosol types are significantly modified when they co-exist. Therefore, it would seem better to refer to the change in radiative forcing as a dust-pollution interaction. The fact that the dust was there first historically is besides the point. However, perhaps this is a matter of interpretation that will be difficult to agree on. In that case perhaps a few minor edits will rectify this. For instance: Line 4 of the abstract. "This results in an anthropogenic radiative forcing associated with mineral dust..." Can I suggest: "This results in a change in aerosol direct radiative effect that we interpret as an anthropogenic radiative forcing associated with mineral dust-pollution interactions... Or: "This results in a change in aerosol direct radiative effects that we interpret as an anthropogenic radiative forcing associated with mineral dust, although it includes impacts of the dust on the pollution."

We have adopted "This results in a change in the aerosol direct radiative effect that we interpret as an anthropogenic radiative forcing associated with mineral dust-pollution interactions."

> Likewise in the second sentence of the conclusions. Please could the authors add something like: "We interpret this as..."

We have replaced "This causes..." by "We interpret the resulting effect on the radiative transfer as".

> Other minor suggestions: 1. Thanks for explaining how figures 1 and 2 were calculated (i.e. using equation 1). This helps me but is this now noted this in the text?

We have added "(based on Eq. (1))" to the figure captions of both figures (so far the use of Eq. (1) for AOD and burdens was only indicated below Eq. (1)).

> 2. It is great to see the changes in dust, BC, seasalt, and water in Figure S11. Would it is be possible to include OC? I think this would complete the picture.

To complete the picture despite the lack of model output for the OC concentrations we have added Fig. S21 which is closely related to Fig. S11 and includes OC as part of the organic aerosols, showing the contribution of the different components to the AOD at 550 nm (dominated by the accumulation mode) and at 10 um wavelength (dominated by the coarse mode).

> 3. Figure 6. It would still be interesting to see the bar for the pollution forcing to put the other bars in context.

We have added the bar for the pollution forcing.

> Otherwise, the manuscript is in good shape and will be an interesting article for ACP and a valuable contribution to understanding the interplay between natural and anthropogenic aerosol within the Climate System.

[revised manuscript text omitted]